# Effect of Systemic Inflammation in the CNS: A Silent History of Neuronal Damage

**DOI:** 10.3390/ijms241511902

**Published:** 2023-07-25

**Authors:** Mara Verónica Millán Solano, Citlaltepetl Salinas Lara, Carlos Sánchez-Garibay, Luis O. Soto-Rojas, Itzel Escobedo-Ávila, Martha Lilia Tena-Suck, Rocío Ortíz-Butrón, José Alberto Choreño-Parra, José Pablo Romero-López, María Estela Meléndez Camargo

**Affiliations:** 1Red MEDICI, Carrera Médico Cirujano, Facultad de Estudios Superiores Iztacala, Universidad Nacional Autónoma de Mexico, Tlalnepantla 54090, Mexico; mara.vsm@gmail.com (M.V.M.S.); carlos.s.garibay@hotmail.com.mx (C.S.-G.); oskarsoto123@unam.mx (L.O.S.-R.); iescobed@ifc.unam.mx (I.E.-Á.); pabloromero@iztacala.unam.mx (J.P.R.-L.); 2Laboratory of Immunobiology and Genetics, Instituto Nacional de Enfermedades Respiratorias Ismael Cos’ıo Villegas, Mexico City 14080, Mexico; choreprr@gmail.com; 3Departamento de Neuropatología, Instituto Nacional de Neurología y Neurocirugía Manuel Velasco Suarez, Mexico City 14269, Mexico; mltenasuck@gmail.com; 4Laboratorio de Patogénesis Molecular, Laboratorio 4, Edificio A4, Carrera Médico Cirujano, Facultad de Estudios Superiores Iztacala, Universidad Nacional Autónoma de México, Tlalnepantla 54090, Mexico; 5Departamento de Neurodesarrollo y Fisiología, Instituto de Fisiología Celular, Universidad Nacional Autonoma de Mexico, Mexico City 04510, Mexico; 6Laboratorio de Neurobiología, Departamento de Fisiología de ENCB, Instituto Politécnico Nacional, Mexico City 07738, Mexico; mortizb@ipn.mx; 7Laboratorio de Farmacología, Departamento de Farmacia, Escuela Nacional de Ciencias Biológicas, Instituto Politécnico Nacional, Av. Wilfrido Massieu Esq. Manuel Luis Stampa S/N, U.P. Adolfo López Mateos, Mexico City 07738, Mexico; emelendezc@hotmail.com

**Keywords:** neuroinflammation, pathophysiology, neuroimmunology, systemic inflammation

## Abstract

Central nervous system (CNS) infections including meningitis and encephalitis, resulting from the blood-borne spread of specific microorganisms, provoke nervous tissue damage due to the inflammatory process. Moreover, different pathologies such as sepsis can generate systemic inflammation. Bacterial lipopolysaccharide (LPS) induces the release of inflammatory mediators and damage molecules, which are then released into the bloodstream and can interact with structures such as the CNS, thus modifying the blood–brain barrier’s (BBB´s) and blood–cerebrospinal fluid barrier´s (BCSFB´s) function and inducing aseptic neuroinflammation. During neuroinflammation, the participation of glial cells (astrocytes, microglia, and oligodendrocytes) plays an important role. They release cytokines, chemokines, reactive oxygen species, nitrogen species, peptides, and even excitatory amino acids that lead to neuronal damage. The neurons undergo morphological and functional changes that could initiate functional alterations to neurodegenerative processes. The present work aims to explain these processes and the pathophysiological interactions involved in CNS damage in the absence of microbes or inflammatory cells.

## 1. Introduction

The innate immune system response to tissue damage is called inflammation, and its principal objectives are eliminating, limiting, and controlling the origin of the damage [1]. During this response, cells and signaling molecules are involved. An efficient inflammatory response is able to eliminate or neutralize microorganisms or toxins. If the response is insufficient, it perpetuates and generates an excessive release of endogenous and exogenous mediators, affecting distant organs from the initial focus and leading to systemic inflammation.

Sepsis was redefined in 2016 (“Sepsis-3”) as “a life-threatening organ dysfunction caused by a dysregulated host response to infection” [2]. The severity is determined by the presence or absence of organ dysfunction [2,3]. The target organ of sepsis is the endothelium, which initiates the inflammatory response and perpetuates damage.

The central nervous system (CNS) is considered an immune-privileged tissue and is believed to be unable to initiate an inflammatory immune response in the presence of pathogens. This property is attributed to the presence of barriers and the absence of lymphoid organs [4,5]. However, the current knowledge indicates that the CNS can detect systemic inflammation through three pathways: (1) neural, vagus, trigeminal, and olfactory nerves, which are activated by pathogen-associated molecular patterns (PAMPs) and inflammatory cytokines; (2) humoral, orchestrated by circulating inflammatory mediators that reach the CNS via the choroid plexus and circumventricular organs; and (3) through the CNS barriers, in which activated endothelial cells release inflammatory mediators [6]. The BBB is present throughout the brain microvasculature. The BBB is formed by the endothelium that possesses tight junctions, which confer selectivity to the passage of some elements. The blood–cerebrospinal fluid barrier (BCSFB) is localized in the lateral, third, and fourth ventricles. These barriers possess high concentrations of antioxidant enzymes that protect against oxidative stress [7]. In the CNS, there are structures within these barriers, and these structures are called the circumventricular organs. Recent work has described the presence of neuroinflammation in the absence of injuries or direct damage to the CNS, but in the presence of systemic inflammatory processes, where the participation of CNS resident cells plays a role in the initial response and progression of the phenomenon [8]. Neuroinflammation is an innate immune response of nervous tissue to restrain infections and eliminate pathogens and cellular detritus [9]. Microglial cells are the primary initiators of brain immune responses; however, both microglia and astrocytes are activated and contribute to the acute phase of inflammation [10,11], therefore protecting the neurons from damage. If the response is not regulated, it causes an alteration in the structure and function of CNS cells [12], which can cause neurodegeneration [13]. Neurodegeneration is observed after injury and in conditions related to the development of neurodegenerative diseases that negatively impact the health of individuals [13,14]. The present work aims to explain the physiopathological mechanisms of neuroinflammation in processes lacking a direct infection in the CNS (Figure 1).

## 2. Local Infections Generate Systemic Inflammatory Response Syndrome

Sepsis is defined as an organic dysfunction caused by a dysregulated response to infection. One of the most common sites that generates sepsis is the lungs, followed by the digestive tract [15].

During systemic inflammation, exogenous and endogenous mediators are released. These exogenous mediators can initiate and generate an immune system response. One of these exogenous mediators is LPS, which is recognized as a PAMP, and is a structural component of the cell wall of Gram-negative bacteria. When LPS is administered systemically, it induces an exacerbated inflammatory response [16]. LPSs can bind to pattern recognition receptors (PRR) and cell surface molecules in cells such as on monocytes, macrophages, and granulocytes and with lower intensity to B lymphocytes, such as CD14 and LPS-binding protein (LBP), which form a complex that favors sensitivity to LPSs and affects a functional signal on cells [17]. The result of this LPS-CD14 receptor interaction enhances the phosphorylation and activation of intracellular kinases, protein tyrosine kinases (PTKs), and mitogen-activated protein kinases (MAPKs) [18,19]. Activated kinases modulate gene expression by increasing the transcription of genes encoding proteins such as cytokines, cytokine receptors, adhesion molecules and NO (nitric oxide) synthase enzymes, and acute phase proteins (complement, reactive C protein) [16,20,21].

Endogenous mediators released at the serum level during this phenomenon include complement (C3a and C5a) and arachidonic acid metabolites such as prostaglandins (PG), thromboxanes (TX), and leukotrienes (LT). Other endogenous mediators that are released are proinflammatory cytokines such as tumor necrosis factor α (TNFα), interferon γ (IFNγ), interleukin 1 (IL-1), interleukin 6 (IL-6), C-X-C motif chemokine ligand 10 (CXCL10) or interferon gamma-induced protein 10 (IP-10), and C-reactive protein (CRP) [21,22,23,24]. In addition, anti-inflammatory cytokines such as IL-4, IL-10, IL-13, or transforming growth factor beta (TGF-β) all regulate the function of pro-inflammatory cytokines. The release of free radicals such as reactive oxygen species (ROS) and nitric oxide species (NOS) by LPS-activated neutrophils, monocytes, macrophages, and endothelial cells contributes to the phenomenon of tissue damage [25].

In addition to the role of sepsis increasing the serum levels of cytokines, danger signals or alarmins, other pathologies such as pancreatitis, major surgeries, burns, and chronic inflammatory states such as ulcerative colitis, obesity, type 2 diabetes, cancer, and autoimmune diseases can induce an increase in the levels of other serum molecules such as the danger-associated molecule calcium-binding protein S100A8 (S100A8) and high mobility group 1 proteins (HMGB1) [26,27,28].

S100A8 is an inflammation-related protein secreted mainly by activated neutrophils and macrophages during acute inflammation [26,29,30]. Circulating S100A8/A9 levels are also increased in hospitalized patients with septic shock, and sepsis survivors show lower serum S100A8/A9 levels than no survivors, which is considered a predictor of mortality in patients with septic shock [31,32].

HMGB1 is a nonhistone nuclear protein that, under normal conditions, is in the nucleus and can establish a protein–protein interaction with the specific transcription factors p53 and p73, and nuclear factor κB (NF-kB), which are present in almost all cells that regulate the activity of the innate immune system [33,34]. This molecule can be secreted actively or passively and can function as damage-associated molecular patterning molecules (DAMPs) to mediate inflammation and immune responses [35,36].

In a clinical study that involved 64 patients with septic shock and severe sepsis, Sudén-Cullberg et al. studied the kinetics of the HMGB1 molecule during the disease course. This molecule was elevated in 34 patients with severe sepsis and septic shock, and the serum concentration at 144 h, the last sampling point, was 300 times higher than any of the other cytokines evaluated during the study [37]. In another clinical study, a similar increase in plasma HMGB1 levels was observed; however, this demonstrated an association between the elevation of this molecule and sepsis severity and mortality [38]. The dysregulated, uncontrolled release of these molecules into the bloodstream perpetuates damage and affects organs distant from the initial infectious focus, such as the CNS [39,40,41].

However, the inflammatory effect at the blood level has been limited by the presence of the “barriers” of endothelial cells; nevertheless, endothelial cells in their active form can trigger a cascade of interstitial and cellular events.

## 3. Endothelial Damage

During systemic inflammation, the endothelium and the coagulation pathway are activated. Thrombin, which is the regulator of the coagulation pathway, is activated from prothrombin by Factor X. Thrombin converts fibrinogen to fibrin and activates platelets, forming microthrombi in capillaries and generating hypoperfusion and metabolic failure in tissues [42,43,44].

On the other hand, anticoagulant agents are downregulated, inducible nitric oxide enzyme activity increases, and pathological vasodilatation appears in general circulation. Capillary vasoconstriction and tissue hypoperfusion occur. In epithelial cells, alterations occur in Na/K/ATPase channels, and an electrolyte concentration generates intracellular edema. These events cause tissue injury and fluid movement from the intravascular space to the interstitial space [45].

The endothelium of blood vessels is composed of a monolayer of endothelial cells resting in a basement membrane, which has a glycocalyx, which is a surface layer consisting of glycoproteins, proteoglycans, and entrapped plasma proteins [46,47]. This structure regulates vascular permeability, preventing the passage of different molecules through the endothelium, depending on their size and charge.

The release of LPSs, cytokines, and mediators of the coagulation cascade during sepsis can alter the thickness and stiffness of the endothelial glycocalyx, increasing vascular permeability. An in vitro study on endothelial HPMECs cells demonstrated that the glycocalyx is sensitive to LPS treatment, which reduces the thickness and stiffness of the glycocalyx by 48% and 29%, respectively, compared to the control. The exposure of these cells to thrombin or TNFα reduced the glycocalyx thickness from 48% to 55% and stiffness from 36% to 35%. The endothelial glycocalyx is susceptible to classical mediators secreted during sepsis, causing alterations in blood vessel permeability [48]. Signs of ischemia have been observed in several brain areas as a response to endothelial damage during sepsis. Some of the most affected areas of the CNS are the hippocampus, amygdala, frontal cortex, and hypothalamus [49,50,51].

The vascular structures are not the same in all organs, as they depend on the latter’s functions and interact with the cells; thus, fenestrated endothelia, sinusoidal types, and tight junctions characterize the BBB.

## 4. CNS and Systemic Inflammation

CNS is protected against harmful substances in the blood by the BBB and the BCSFB [52,53,54]. However, systemic inflammation impairs the brain parenchyma, resulting in three processes: neuroinflammation, ischemia, and cellular metabolic stress [49,50,51]. The presence of these processes demonstrates that there is some damage to the barriers protecting the CNS [49].

The BBB is a highly selective and dynamic interface between the brain parenchyma and the peripheral circulation. It consists of brain endothelial cells, pericytes, and astrocytes [55,56,57]. Under normal conditions, the BBB serves as a physical barrier. Endothelial cells possess tight junctions that restrict the diffusion of molecules between them, forcing most molecular traffic to take a controlled transcellular route through them [58].

In sepsis, these cytokines (IL-1α, IL-1β, IL-6, and TNFα) can be released in the blood and cross the BBB through their respective transporters [59,60]. LPS can also be released during sepsis, can interact with elements of the blood–brain barrier, increase permeability, and activate the glia [61,62,63,64,65]. In murine models of pulmonary infections by Gram-negative bacteria such as Klebsiella pneumoniae, by Gram-positive bacteria such as Mycobacterium tuberculosis, or by H1N1 influenza virus, infected mice generate neuroinflammation without direct infection in the brain, with elevated levels of IL-1β, TNFα, keratinocyte chemoattractant (KC), IL-12, IL-17, IP-10/CXCL10, DAMP S100A8, monocyte chemoattractant protein-1 (MCP1), IFN-γ, iNOS, and 2,3-dioxygenase (IDO). There is also increased vascular permeability, myeloid cell infiltration, and structural alterations in the neurons. These models have shown that systemic inflammation generates alterations in animal behavior [41,66,67]. In humans, clinical symptoms, such as altered mental status, cognition, sleep patterns, and thinking, are expressed due to sepsis-induced neuroinflammation in patients with sepsis-associated encephalopathy syndrome. Other warning signs are agitation, hallucinations, and coma [68,69,70].

A clinical study demonstrated the activation of astrocytes glial fibrillary acidic protein (GFAP) and macrophages (CD68/CD45) and the elevation of chemokines (CXCL8/10/12, CCL13/22) in the encephalon of patients diagnosed with sepsis. The elevation of these chemokines is related, on the one hand, to the activation of a proinflammatory environment and, on the other hand, to the chemoattraction of myeloid cells in the brain parenchyma, exacerbating the phenomenon of neuroinflammation and conditioning to cognitive and functional impairment in these patients [71].

BCSFB is in the choroid plexus and the brain ventricles. The choroid plexus is composed of epithelial–endothelial cells, a stroma with highly vascularized connective tissue and epithelial cells [72,73,74,75]. The stroma consists of fenestrated capillaries, connective tissue, and immune cells. The ventricular side of the stroma is shielded by a monolayer of cuboidal epithelial cells [73,74]. Epiblast cells or Kolmer cells show phagocytic activity and adhere to the ventricular portion of the epithelial cells [76,77].

The BCSFB barrier plays a crucial role in systemic-inflammation-induced neuroinflammation [78,79,80,81]. Sánchez-Garibay et al. demonstrated that a pulmonary infection with certain strains of M. tuberculosis can induce changes in brain barriers and specifically alter the integrity of the BCSFB [82].

In murine models of sepsis, the presence of LPS at the systemic level induces the upregulation of Toll-like receptors (TLR1, TLR2, TLR3, TLR4), CD14, cyclooxygenase-2 (COX-2), nuclear factor of kappa light polypeptide gene enhancer in B cells inhibitor, and alpha (IκBα) in choroid plexus cells, which can lead to the decreased expression of occludin in the CSBFB [83,84,85,86,87,88,89].

During local neuroinflammation, CSBFB damages the endothelium, increasing the permeability of this structure [90]. In a clinical study, 606 patients out of 823 with neuroinflammation presented an abnormal elevation of the sCD146 molecule in the cerebrospinal fluid (CSF). The elevation of this molecule is related to the increased expression of matrix metalloproteinase-2 and 9 (MMP2 and MMP9), intercellular adhesion molecule (ICAM), vascular cell adhesion molecule (VCAM), IL-1β, IL-17, and IFNƔ in CSF. In vitro experiments demonstrated that the effect of this molecule on endothelial hCMEC/D3 cells causes a reduction in the expression of zonula occludens-1 (ZO-1), junctional adhesion molecules (JAM), and occludin and interacts with integrin β1, alters the cytoskeleton, increases caspase 3/9, decreases Bcl-2 (anti-apoptosis), and increases Bax (pro-apoptosis). The signaling pathway used by CD146- integrin αvβ1 involved MAPK, extracellular signal-regulated kinases 1 and 2 (ERK1/2), c-Jun N-terminal kinases (JNK), p38, serine-threonine protein kinase (Akt), and NF-kB [90]. NF-κB activation regulates the overproduction of MMP-9 and induces claudin-5 degradation, leading to altered CSBFB integrity [91].

In a murine model during cecal ligation puncture (CLP)-induced systemic inflammation, it was demonstrated that oxidative stress plays an important role in the choroid plexus, increasing BCSFB permeability and leading to alterations in CNS function. Oxidative stress leads to altered function in the electron transport chain of CNS-resident cell mitochondria, as well as an increase in TBA (thiobarbituric acid) resulting from lipid lipoperoxidation, decreased catalase, superoxide dismutase (SOD) and ROS, O_2_, OH, and H_2_O_2_ in the cerebrospinal fluid [92,93].

In other murine models and clinical cases of sepsis, the serum upregulation of IL-6, TNFα, and HMGB1 was observed [31,37,39,94]. The release of these elements causes neuroinflammation. HMGB1 can bind to the receptor for advanced glycation end products (RAGE) and TLR4 receptors on astrocytes and microglia in the hypothalamus. The binding of this molecule to these receptors initiates intracellular signaling, activates NF-kB, and promotes the transcription of genes that encode proteins such as IL-6, a cytokine related to neuroinflammation. HMGB1 could be one of the causes of the damage since it can initiate and perpetuate neuroinflammation during sepsis without direct injury to the CNS [39,94] (Figure 2).

The BBB gives the cerebral vasculature the previously mentioned properties. In the brain, some structures do not have a BBB; the circumventricular organs (CVOs) and their locations are strategic due to their proximity to neuroendocrine and autonomic centers. The CVO can perceive stimuli from the bloodstream and express receptors for inflammatory cytokines and bacterial fragments [95,96].

## 5. CVO and Systemic Inflammation

HMGB1 protein can be released into the extracellular space in two ways: actively, by immune cells in response to an inflammatory stimulus since it is a cytokine; and passively, by cells undergoing necrosis and releasing intracellular proteins such as HMGB1 [36,97].

In neurons of the area postrema (AP) and area preoptic nucleus (MnPO) of the circumventricular organs, HMGB1 is normally found in the cytoplasm and perinuclear regions. In vitro, it was shown that neurons in the AP stimulated with LPS and HMGB1 induced an increase in extracellular HMGB1 and a decrease in intracellular HMGB1R. The release of HMGB1 into the extracellular space causes an interaction with RAGE and TLR4 receptors, activating internal signaling pathways that promote the synthesis of inflammatory molecules and exacerbating neuroinflammation [39,98].

Microglia and perivascular macrophages of the vascular organ of lamina terminalis (OVLT), subfornical organ (SFO), median eminence (ME), and AP activate the NF-kB p65 pathway upon TLR2 receptor stimulation with LPS and Pam3CSK4 [99,100]. The interactions of circulating elements form the systemic inflammatory response in these structures and initiate a neuroinflammatory response.

## 6. CNS Endothelium and Systemic Inflammation

As already reviewed, the CNS’s anatomical and physiological characteristics are different from many other organs; one example is the BBB, which confers physicochemical characteristics that result in selectivity to the passage of ions, nutrients, oxygen, and water. Its endothelium has specific characteristics different to the rest of the endothelium. The presence of tight junctions is the main difference, since this structure, formed by a protein complex, confers selectivity to the passage of circulating elements in the bloodstream. The protein complex consists of individual elements such as occludin, claudins, ZO-1, JAM-A, JAM-B, and JAM-C [55,101].

During systemic inflammation induced by LPS, alterations are made at the cellular level and affect the function of this barrier, increasing permeability and myeloid cell infiltration [102]. In a clinical study with 22 patients diagnosed with deadly sepsis, ZO-1 protein was absent in the brain in all cases, and claudin-5 was only present in 4% of the cases. Occludin, which is a protein that maintains the integrity of tight junctions, was not expressed in 38% of the cases (18 patients). This indicated endothelial damage to the barrier during sepsis. The absence of occludin in these patients correlates with elevated C-reactive protein (CRP) and procalcitonin (PCT) levels [15].

In sepsis, hypoperfusion occurs at the level of the CNS capillaries, leading the cells to a state of glucose and oxygen deprivation. Under these conditions, there is an increase in ROS due to the depletion of glutathione (GSH) in endothelial cells. The increased ROS induces a continuous disorganization of claudin, occludin, and β-catenin in endothelial cells and increased BBB permeability [103].

LPS can activate the COX2/PGE2/NF-kB signaling pathway in brain endothelial cells [61,62,104,105]. The activation of this signaling pathway causes a decrease in ZO-1 at endothelial cell borders, giving a hazy appearance to the cytoplasm and causing an increase in endothelial permeability, paracellularly, and translucency [62]. Endothelial cell NF-kB activation induces proinflammatory cytokine synthesis and polymerase delta-interacting protein 2 (Poldip2) expression, which induces the p65 translocation of NF-kB [61].

On the other hand, in murine models, it has been shown that LPS can generate increased CXCR2 (chemoattractant receptor) expression and neutrophil infiltration in addition to the rolling and adhesion of leukocytes in the microvasculature of the central nervous system [106]. One of the mechanisms by which cell infiltration can be explained was demonstrated by an in vitro model in which bEND3 endothelial cells were stimulated with CXCL1 (endogenous ligand of CXCR2). This molecule induced the synthesis of actin stress fibers, damaged tight junctions, and increased permeability, thus contributing to neuroinflammation [106]. When rBMECs are stimulated with elements of the systemic inflammatory response (D-lactate, TNFα, IL-6, IL-8), they induce miR-181b expression [107]. miR-181b is an inflammation- and apoptosis-associated gene. The JAK2/STAT signaling pathway induces miR-181b expression and promotes endothelial cell damage. On the other hand, in murine studies of the cecal ligation and puncture (CLP) sepsis model, the overexpression of miR-181b was shown to attenuate neuroinflammation through the downregulation of NF-ΚB, IL-1β, and TNFα. Nevertheless, this response has only been observed in the hippocampus [108].

Moreover, IFN-γ is released during the systemic inflammatory response into the bloodstream; it binds to endothelial cells at the BBB and activates the MyD88 signaling pathway, which, in turn, induces the expression of CXCL10, a molecule associated with myeloid cell recruitment, and lipocalin-2 (Lcn2) [109]. Lnc2 is a key regulator of MMP function, and the impact of this molecule on the basement membrane is fundamental to maintaining its integrity, causing an increase in permeability, cell infiltration, and neuroinflammation. In this way, it can be explained that in these animal models, a deterioration of behavior and memory is observed. On the other hand, it has been shown that Lcn2 is released into the plasma in response to ischemia [110].

Pericytes are another important cell type in the BBB, and these cells are intercalated in the basement membrane of the BBB, whose function is to preserve the integrity of vessels and regulate capillary blood flow in the central nervous system. Pericytes also regulate astrocyte end-foot polarization at the BBB [111], and it is now known that aquaporin 4 (AQP4), the main water channel present in astrocyte end-feet, is also part of pericyte binding [112].

CNS pericytes express diverse receptors for cytokines, pathogens, and molecules such as TNF-α, IL-1β, IL-8, LPS, and α -synuclein. The binding of these molecules to their respective receptors induces the synthesis of inflammatory molecules [63,69,113,114,115].

A murine model of septic encephalopathy demonstrated that LPS generates basal lamina disruption and the detachment of lamina pericytes, altering function and increasing barrier permeability [63].

Inflammatory cells are usually associated with macrophages, neutrophils, eosinophils, lymphocytes, mast cells, etc. These cells, cytokines, factors, hormones, and peptides play a role and participate in almost all processes. In an intact brain, these cells hardly cross the blood–brain barrier and interact with glial cells. Previously, it was believed that this was entirely necessary; however, glial cells are a whole active neuroimmunological system of their own (Figure 2).

## 7. Glial Cells and Systemic Inflammation

Astrocytes are the most abundant cell type in the CNS; their function is to regulate the environment of neurons, maintain the homeostasis of the microenvironment, and regulate the concentrations of ions, cations, neurotransmitters, amino acids, and glucose [116,117,118,119].

LPS-stimulated astrocytes show a profile of the classical proinflammatory pathway and induce the expression of genes encoding proinflammatory cytokines and chemokines such as IL-1, iNOS, TNF-α, CXCL10, and Lcn2 [120]. Lcn2 is produced by astrocytes stimulated by LPS, and this molecule has a paracrine effect on nearby astrocytes, showing a classical proinflammatory profile.

The induction of A1 reactive astrocytes was demonstrated in an in vitro neuron-astrocyte coculture model. Astrocytes were cocultured with neurons in oxygen–glucose deprivation (OGD), and this altered the expression/distribution and activity of glial glutamate transporters, impaired cellular glutamate uptake, and decreased intracellular levels of glutathione in neurons, decreasing the viability of the neurons. These results question the paradoxical role of astrocytes in a proinflammatory profile and their classically described function [121].

In sepsis, hypoperfusion occurs at the level of CNS capillaries, leading cells to a state of glucose and oxygen deprivation and stimulating GSH activity. An in vitro study demonstrated that GSH is transferred from astrocytes to endothelial cells under both normoxic levels and hypoxic and glucose deprivation conditions [103]. GSH efflux from astrocytes to endothelial cells is paracellular to protect the endothelium from damage from hypoxia and glucose deprivation. In contrast, under these same conditions, astrocytes can express VEGF-A and MMP-9 through the JAK2/STAT3 signaling pathway. These molecules are related to the ischemia-induced loss of BBB integrity [122].

In the brain parenchyma, although astrocytes are indeed found in higher numbers, like glial cells, in addition to oligodendrocytes, other cells are derived from the hematopoietic precursors of the embryonic stage and share this lineage with monocytes. These cells were first described by Del Río Hortega as microglia [123]. Microglia are the resident phagocytic cells of the CNS, corresponding to only 10% of the cell population. Its function includes the elimination of foreign elements from the CNS, and it is the main effector of the inflammatory response at the central level [124,125]. These cells can be activated by the humoral pathway or neural pathway, or by alterations in the BBB once activated microglia can release cytokine chemokines and reactive oxygen and nitrogen species [6,126,127].

Remarkably, when stimulated by LPS, microglia increase CD40 expression in the hippocampus. Consequently, this increases the expression of CD40 and CD40L, which induces the release of the proinflammatory cytokines TNF-α, IL-1β, and IL-6 and causes increased levels of reactive oxygen and nitrogen species, increasing inflammation in the hippocampus [128]. Microglia activated by the CD40/CD40L pathway upon LPS stimulation regulate the inflammatory response, myeloid cell infiltration, and expression of inflammatory molecules that cause an increase in barrier permeability.

Similarly, it has been demonstrated in murine models that microglia, when stimulated with LPS, activate their phenotype of the classical M1 or neurotoxic pathway and express genes related to this profile, such as iNOS, TNF-α, CXCL10, IL-12, IL-23, and Lcn2 [129]. This last molecule, also called neutrophil gelatinase-associated lidocaine, is involved in immunity by sequestering iron and preventing its use by bacteria. It has a paracrine effect on nearby microglia in M1 polarization, increasing the inflammatory response. LCN2 also inhibits STAT6 activation and M2 gene expression in microglia along with the phagocytic activity of M2 microglia stimulated by IL-10 [129].

LPS generates a proinflammatory response by microglia and induces an increase in MMP 2, 9, and 13 expression through the activation of the PI3K signaling pathway [64]. Metalloproteins (MMPs) are involved in injury to the basement membrane of the BBB and cause the reorganization and degradation of tight junction proteins [130,131], causing increased permeability and the infiltration of myeloid cells, increasing neuroinflammation. However, as mentioned above, in situations of systemic inflammation, the S100A8/A9 molecule can be released in the blood [28] and can also be secreted by neurons under hypoxic conditions [132]. Microglia stimulated with LPS is also able to release S100A8 [40]. This molecule is key for the activation of neuroinflammation without direct CNS damage, coupling to RAGE, TLR4, and NLRP3 receptors, activating the ERK/JNK/NF-kB signaling pathway of microglia. It also generates a morphological change and increases the expression of iNOS, MMP9, TNF-α, IL-1, CCL2, CCL3, and CXCL10 [40,133,134]. The release of these mediators induces inflammation and the chemoattraction of more inflammatory cells and exacerbates apoptosis in oligodendrocytes. This response may be key to the generation of neurodegenerative pathologies.

Regarding the role of Nod-like receptor protein 3 (NLRP3) inflammasome activation in microglia, NLRP3 can be activated by various stimuli, such as ATP, endoplasmic reticulum stress, LPS, and some cytokines [94,135,136]. The production of interleukin-1β (IL-1β) is primarily dependent on the activation of the inflammasome, a multiprotein complex of cytosolic proteins, such as NLRP3, apoptosis-associated speck-like protein (ASC), and pro-caspase-1 [136]. The activation of inflammasome pathways triggers pyroptosis and leads to the extracellular release of inflammatory cytokines [137]. Pyroptosis is a kind of inflammatory cell death regulation that relies on the activation of cytosolic inflammasomes, mainly depending on the caspase family [138].

IFN-γ, which is released during systemic inflammation, couples to and activates the TLR4 receptor, inducing a proinflammatory phenotype of hippocampal lba1, CD11b, and CD68 microglia [139]. Lipoteichoic acid (LTA), a major component of the cell wall of Gram-positive bacteria, can couple to the TLR2 receptor of microglia and generate the same effect [140,141].

LPS, LTA, and IFN-γ cause the elevation of the proinflammatory cytokines IL-1β, TNFα, and IL-6 and the overexpression of iNOS in microglia, exacerbating neuroinflammation [139,140,141,142].

In a murine model with MRL/lpr mice treated with LPS intraperitoneally, neuroinflammation was exacerbated. In the first step, microglia migrate close to the vessels to protect the integrity of the barrier and then change to their reactive form expressing CD68, CCL5, and CLDN5. These last two molecules are part of the endothelial tight junctions. Active microglia invade and infiltrate the basement membrane, interacting with the endothelium of the BBB and increasing the permeability of the barrier. When stimulated with IFN-γ, the microglia migrate near vessels and elevate CCL5 (which is part of the endothelial cell junction) and CLDN5 (which promotes infiltration to the neurovascular unit (NVU), interacting with and maintaining BBB integrity [65].

Oligodendrocytes are cells responsible for axonal myelination at the CNS level. In vitro models of oligodendrocyte precursor cells (OPCs) cocultured with LPS-stimulated microglia or astrocytes can cause a decreased viability of oligodendrocytes. The damage to OPCs could occur through susceptibility to LPS or through LPS-activated glia [134,143,144].

In EP1-/- transgenic mice and purified OPCs, Carlson demonstrated that PGE2 directly inhibits OPC maturation in an EP1-dependent manner. Pharmacological blockade with specific EP1 inhibitors (ONO-8711 or SC-51089) also attenuated the inhibitory effects of PGE2 on OPC maturation. In vitro models showed that PGE2 blockade attenuates oligodendrocyte damage [145]. COX2 is not only a marker of “A1” reactivity, but may also function to promote the arrest of OPC maturation through PGE2 production [145,146].

Other studies have shown that LPS or hypoxic ischemia causes fewer mature oligodendrocytes and alterations in their function, resulting in a hypomyelination phenomenon in neurons [147,148]. The synergistic effect of exposure to systemic inflammatory response products causes damage to oligodendrocyte precursor cells conditioned to neuronal damage.

In physiological processes, there is a balance between elements of the BBB (endothelial cells, pericytes, astrocytes), glial cells, and neurons. In a microenvironment, intercellular communication through peptides, neurotransmitters, cytokines, chemokines, ions, and other factors orchestrates all known neurophysiological processes. In the CNS, the neurons, which are the basic functional unit, unlike the rest of the tissues where there is a repair process by cell regeneration after a tissue injury, do not have this capacity to repair the damage caused by injury. All protective mechanisms are brought into play to protect neurons from damage during systemic inflammation (Figure 3).

## 8. Neurons and Systemic Inflammation

During and after the process of systemic inflammation by LPS, hippocampal and substantia nigra neurons undergo autophagy, necrosis, and apoptosis [67,149,150,151,152], and some other neurons are susceptible to pyroptosis [153,154]. Since the number of mitochondria decreases, the nuclear and mitochondrial membranes show interruptions in the continuity of the double neural nuclear membrane. In murine models of cecal ligation and puncture (CLP) or LPS sepsis, neuronal death by ferroptosis has been observed. These studies have shown that the mitochondria contract and transferrin, NADPH oxidase 1 (NOX1), malondialdehyde (MDA), and neuron-specific enolase (NSE) levels increase; at the same time, there is a reduction in proteins such as nuclear factor 2 (Nrf2), related factor E2, and glutathione peroxidase 4 (GPX4)-dependent antioxidant enzymes [155,156]. Ferroptosis is a programmed cell death characterized by iron accumulation, ROS production, and lipoperoxidation, leading to an inflammatory cascade and glutamate release [157,158,159]. Given these results, sepsis can generate excitotoxic neuronal injury related to glutamate release.

Similarly, neuron death of the amygdala, hypothalamus, and medulla in postmortem brain tissues of patients who died from sepsis has been observed [50,160,161,162]. Neuroimaging studies of patients presenting with sepsis concluded that there was a reduction in volume in areas such as the hippocampus and the frontal cortex 6 and 24 months after hospital discharge, thus demonstrating that the damage was perpetuated [163,164,165]. On the other hand, in murine sepsis studies, it has been shown that there are alterations in cognitive functions related to long-term memory [67,149,162,166,167].

The role played by LPS in mitochondrial damage and the presence of oxidative stress has been studied through in vitro model with neurons cells of the SH-SY5Y line when stimulated with LPS. It was substantiated that there is an overproduction in the expression of Aβ1–42 peptide, T-Tau protein, VEGF, and TGF-β and the inflammatory response molecules iNOS, IL-1, IL-6, TNF-α, and ROS. LPS is capable of repressing genes of the mitochondrial complex I subunit (NDUFV1, NDUFV2, NDUFS1, and NDUFAB1), which alters the function of the mitochondria and generates an increase in the rate of oxygen consumption (TCO), leading to an increase in the maximum respiratory capacity and the respiratory reserve capacity. In addition, LPS treatment also affected genes involved in mitochondrial biogenesis, fission, and fusion (PGC-1a, PGC-1b, NRF1, TFAM). Similarly, treatment with LPS altered the actin filaments of the cells [168]. These observations validate the potential role of LPS in amyloidogenesis, tauopathy, and neuroinflammation. The presence of oxidative stress induces neuroinflammation and neurodegeneration [169].

It has been shown that neuroinflammation persists as a chronic and aseptic process, even when there is remission of systemic inflammation due to sepsis [3,170,171,172]. Perivascular IgG deposits, the presence of C1q in the dentate gyrus, and the elevation of TNF-α CCL2, CCL12, CCL17 (CCR2 ligand), CXCL1 (neutrophils), CXCL10 (monocytes), and S100A8 after resolution have been found in some studies [40,41,172,173]. S100A8 elevation is related to long-term functional damage to mitochondria, as ATP levels and oxygen consumption in the hippocampus and prefrontal cortex decrease in acute phases and are maintained days after the induction of systemic inflammation [40]. In long-term cognitive impairment in an experimental model of sepsis, the SOD2 and COX protein content of mitochondria decreased in late phases of damage. Mitochondrial damage during systemic inflammation ranges from edema, a reduction in mitochondrial cristae, alterations in the electron transport chain, and altered ATP production. The function of mitochondrial complexes, such as I and III, causes an increase in the oxygen consumption rate (OCT) [164,174,175]. On the one hand, there is a relationship with mitochondrial complex I dysfunction in α-synucleinopathies [176]. On the other hand, other work has described that the number of mitochondria and mitochondrial cristae decreases in patients with Alzheimer’s disease, and the lack of energy production shifts the metabolism toward beta-amyloid protein precursors [169,177] (Figure 4).

## 9. Clinical Data of Sepsis and Neurological Damage

The clinical manifestations of neurological impairment can be noted in the initial phase of sepsis, at recovery, and long term after hospital discharge. The acute phase of sepsis is characterized by anxiety, anorexia and bodyweight loss, hypersomnia or sleepiness, psychomotor retardation, fatigue, inabilities to calculate and concentrate, and desregulated body temperature [178]. Some other patients present a clinic spectrum called sepsis-associated encephalopathy (SAE), which is defined by changes in consciousness ranging from confusion (delirium) to coma and is characterized by various electrophysiological changes that have been associated with increased mortality and long-term cognitive dysfunction [179,180]. In a prospective observational study with 71 patients with septic shock, they proposed the evaluation of neurological damage during the sepsis episode. The median age of the patients was 65 years (56 to 76) and the results of the neurological examination showed that focal neurological signs were present in 13 (18%) of these patients, seizures in 7 (10%), coma in 33 (46%), and delirium in 35 (49%) [181]. It is true that information about long-term neurological damage in patients who have survived sepsis remains scarce. Some clinical studies showed that neurocognitive dysfunction in sepsis survivors include deficits in spatial memory [182] and impairments in verbal learning and memory [183,184], attention, and vigilance [183].

In a clinic study, 11 subjects were recruited through a follow-up clinic for patients who had been admitted to the ICU for treatment of systemic inflammatory response syndrome 12–18 months after being discharged. To study neurocognitive impairment, a series of cognitive tests was used. Sepsis-recovered patients had significantly impaired performance on pattern recognition memory, impaired spatial recognition memory, and delayed matching to sample tasks. The pattern of impairment observed among sepsis survivors was indicative of possible medial temporal and restricted frontal lobe dysfunction and, more specifically, dysfunction of the parahippocampal complex [182]. These data show that events during the septic episode correlate with long-term cognitive dysfunction.

Despite the recognition of many cellular and molecular recognized elements in sepsis-induced encephalopathy (SAE), there is a lack of consensus regarding the biomarkers or image studies that could benefit early diagnosis. It has been suggested that electroencephalography (EEG) might represent an accessible and effective technique that allows the detection of electrophysiological changes even before the appearance of clinic encephalopathy, although there is still controversy since some studies reported a lack of EEG changes [185]. A study of 33 pediatric patients found EEG alterations in only 26.9% of their cohort; nevertheless, the authors suggest that sedation is an important confounder regarding the study of EEG modifications [186]. Even though most image studies remain unchanged during SAE, recent reports revealed that a volumetric analysis of MRI can detect a loss of gray matter volume that could be associated with a bad outcome [165] and prolonged delirium [164].

Several biomarkers have been proposed to diagnose or manage SAE. Blood neurofilament light chains can be related to neuronal damage [187] and can also predict the development of encephalopathy. Moreover, the CSF levels of NT-proCNP (amino-terminal propeptide of the C-type natriuretic peptide) are increased in SAE patients and remain at high concentrations during the disease course, showing a positive correlation with neuronal damage and inflammatory mediators such as IL-6 and procalcitonin [188]. Finally, several other biomarkers have been proposed, such as plasmatic C-reactive protein, adiponectin, Tau protein, and neopterin [189]. The lack of accurate diagnostic tools makes the clinical management of sepsis-induced neuroinflammation more difficult; nevertheless, future perspectives need to focus on pre-existing methods that would facilitate access in different socioeconomic environments to avoid the long-term neurological impairment of patients surviving sepsis.

## 10. Systemic Inflammation and Neurodegeneration

Growing evidence indicates a strong relationship between neurodegeneration and systemic inflammation [190]. Neurodegenerative diseases are characterized by the progressive, chronic, and irreversible loss of selectively vulnerable neuronal populations [14]. Examples of neurodegenerative diseases are Alzheimer’s disease (AD), Parkinson’s disease (PD), Huntington’s disease (HD), and amyotrophic lateral sclerosis (ALS), among others [191]. These neurodegenerative disorders can be classified according to pathognomonic hallmarks [14,191,192]: (1) primary clinical manifestations, such as dementia, parkinsonism, or motor neuron disease; (2) by the affected neuroanatomical area, such as frontotemporal, extrapyramidal, or spinocerebellar; (3) protein misfolding, such as amyloidosis, tauopathies, α-synucleinopathies, and TDP-43 proteinopathies; (4) neuroinflammation, characterized by the imbalance between the neurotoxic phenotype of M1 microglia/A1 astrocytes and the neuroprotective M2 microglia/A2 astrocytes.

The infectious hypothesis for neurodegenerative disease stipulates the association between these disorders with infectious agents [190]. Interestingly, Chlamydia pneumoniae, Helicobacter pylori, Borrelia Burgdorferi, and Herpes viruses [193] have been found in the brains of AD patients associated with the pathological processing of Aβ, which has been suggested to exert antimicrobial activity. We hypothesized that this systemic inflammation would favor an increase in BBB permeability, a subsequent immune infiltrate that, together with the activation of glial cells, would promote neuronal death.

It has recently been suggested that SARS-CoV2 infection could generate slow and progressive neuronal loss by triggering a neuroimmune response via microglial activation [194], highlighting that microglia can be chronically activated after a single trigger, such as infectious agents [195]. It has also been suggested that SARS-CoV2 could promote a neuroinflammatory environment and the subsequent hyperphosphorylation of tau, a protein involved in AD, promoting the disassembly of microtubules and, subsequently, neuronal degeneration [194,196]. Likewise, this neuroinflammatory response could cause misfolding and aggregation of the α-synuclein protein, involved in the PD pathophysiology [194,197,198]. Therefore, single or repeated infection by SARS-CoV-2 could cause chronic neuroinflammation, favoring protein misfolding and neuronal death. However, future research and cohort studies are required to determine whether SARS-CoV-2 infection is associated with neurodegeneration.

On the other hand, several pathologies and comorbidities such as obesity, diabetes, metabolic syndrome, cardiovascular diseases, and inflammatory bowel diseases (IBDs) have been associated with a chronic inflammatory state, causing a dysfunction in the neural–immune interaction, which could favor the release of proinflammatory mediators within the CNS, culminating in neuronal damage [190,199]. In recent years, the gut-microbiome–brain axis has been the subject of study, since it has been suggested that the intestinal microbiota may have a neuroimmunomodulatory effect through the following mechanisms [199]: (1) direct influence on the neurotransmitters of the enteroendocrine system such as dopamine, serotonin, and catecholamines; (2) mediate signaling between the in the autonomic nervous system and CNS of neuromodulator and neurohormone molecules; (3) modulate the synthesis of short-chain fatty acids and secondary bile acids that stimulate the synthesis and secretion of serotonin from the enterochromaffin cells; (4) immunomodulate the synthesis and release of intestinal pro- and anti-inflammatory molecules. Therefore, the dysfunction of these mechanisms by IBD has been closely associated with various neurological disorders, including neurodegenerative diseases.

Concerning aging, a state of chronic inflammation has been demonstrated that can occur independently of the disease or infection, derived from various physiological changes associated with age, such as cellular senescence and the accumulation of cellular debris. This age-related predisposition to chronic inflammation has been termed “inflammaging”, which has also been linked to neurodegenerative diseases [200].

In summary, single or repeated infections, metabolic disorders, cardiovascular diseases, IBD, as well as aging can cause chronic low-grade inflammation, which, in turn, could trigger an increase in the permeability of the BBB and activation of glial cells, which would cause a neuroinflammatory response that in the long term could lead to the misfolding of proteins such as tau, Aβ, α-synuclein, and TDP-43, all of which are involved in neurodegeneration.

## 11. Conclusions

It has been demonstrated that processes that generate systemic inflammation, such as pancreatitis, major surgeries, extensive burns, sepsis, and some chronic inflammatory processes such as diabetes mellitus, obesity, autoimmune diseases, and cancer, can directly or indirectly damage or alter the nervous system.

For many years, it was asserted that sepsis had no repercussions on the CNS as long as the barrier remained intact; however, today, it is certain that since it is a dynamic, selective structure that is not present in the entire brain, as we have already seen in this review, the opposite happens.

In our review, the objective was to describe the processes that are activated during a systemic event “without apparent damage” to the brain, and the BBB undergoes changes that could allow the passage of LPS, cytokines, and other products of inflammation, triggering a series of local immunological events that lead to damage to neurons.

This damage could be short- and medium-term, where homeostasis mechanisms prevent its clinical manifestation. However, whether it could be plasticity, damage to small neuronal groups, or chronic injury that could trigger the genesis of neurodegenerative diseases has already been documented in many papers. Based on the literature reviewed, we conclude that any systemic inflammatory event triggers neuroinflammation and leads to neuronal damage, as supported by the review.

The question remains as to whether exposure to frequent doses of LPS, trauma, major surgeries, and metabolic diseases such as diabetes mellitus, autoimmune diseases, or cancer would trigger these same processes. If so, it would explain the high prevalence of neurodegenerative diseases.

## Figures and Tables

**Figure 1 ijms-24-11902-f001:**
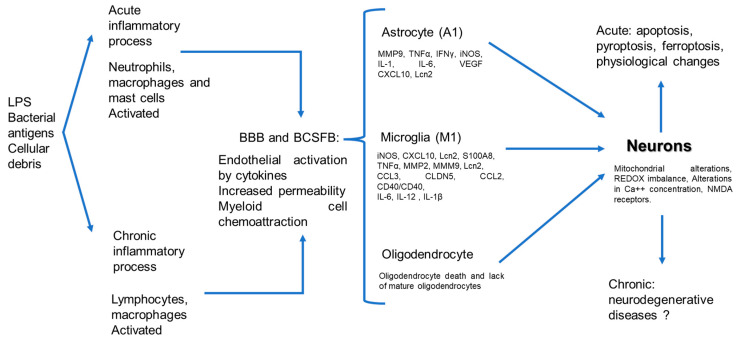
Physiopathogenic process of neuronal damage induced by systemic inflammatory, neuro-immunological pathway.

**Figure 2 ijms-24-11902-f002:**
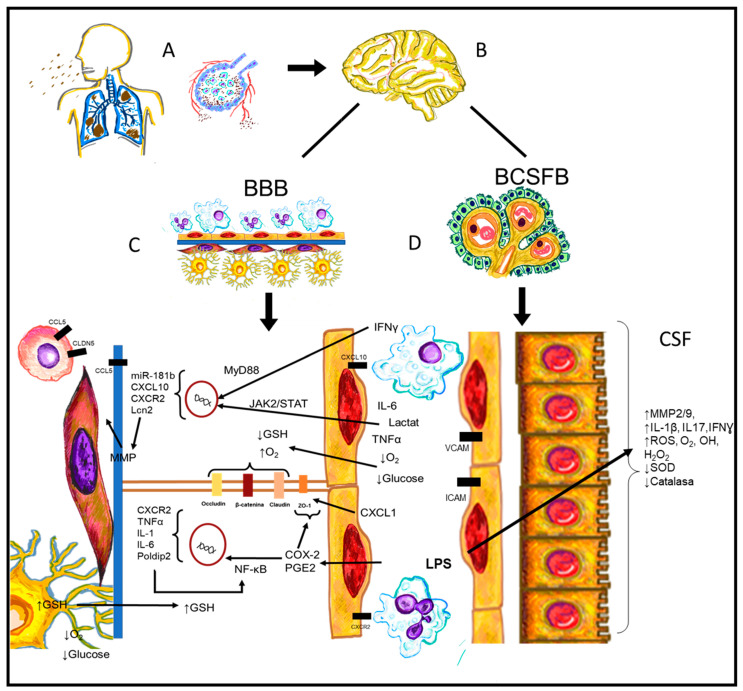
Pathophysiological process from systemic inflammation until the interaction with CNS vascular structures. (**A**) Pulmonary infections such as pneumonia, pleuritis, and bronchitis are released into the bloodstream and spread. (**B**) Systemic infection interacts with CNS, causing encephalitis and/or meningitis. However, antigens such as LPS are sufficient to alter the CNS vasculature. (**C**) BBB suffers not only intracellular molecular changes or alterations, but also structural and intercellular changes of its components. (**D**) BCSFB shows alterations causing endothelial cells to express adhesion molecules and cytokines and alter their cellular structure.

**Figure 3 ijms-24-11902-f003:**
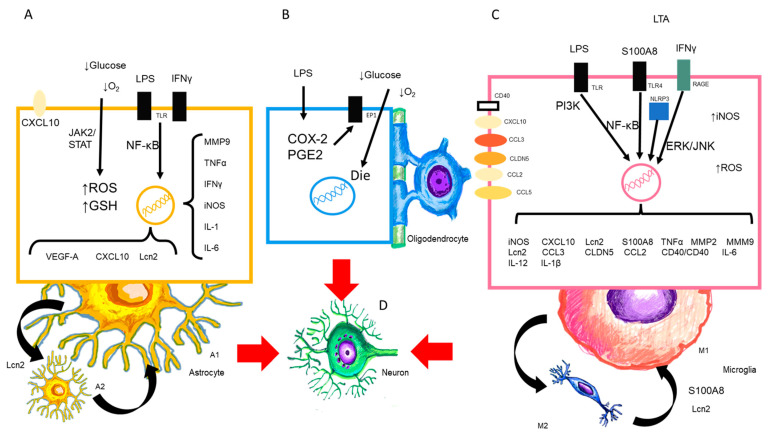
Presence of LPS and CNS cells. (**A**,**B**) LPS activates the astrocyte to its neurotoxic form A1, (**C**) as well as the M1 microglia and the release of cytokines, chemokines, and substances that are released in this process. The oligodendrocyte may also undergo alterations in maturation and death. (**D**) All of the above will target the neuron and cause consequent damage.

**Figure 4 ijms-24-11902-f004:**
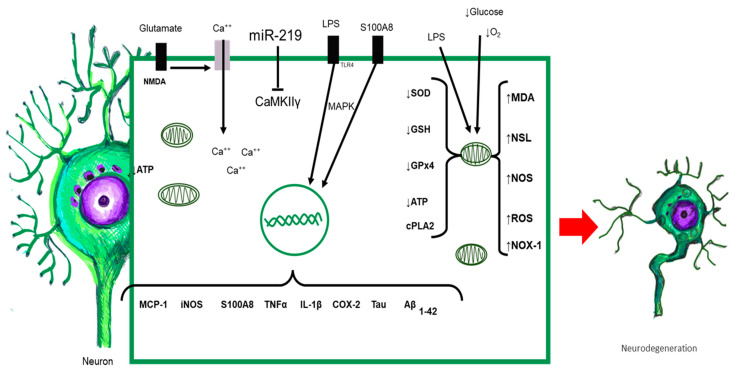
All elements released during sepsis or by activated CNS-resident cells interact with neurons. Neurons possess TLRs capable of being activated upon contact with their ligand, triggering the intracellular signaling cascade, with the synthesis of proinflammatory cytokines, chemokines, and other proteins. Alterations in redox balance and massive activation of NMDA receptors increased intracellular calcium and altered mitochondrial function. The result of all of these responses orchestrated simultaneously leads to morphological and functional alterations of neurons, which could initiate neurodegenerative damage.

## Data Availability

The data that support the findings of this study are available from the corresponding author upon reasonable request.

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
