# Peer review of "Effect of Systemic Inflammation in the CNS: A Silent History of Neuronal Damage"

_ijms, 2023, doi:10.3390/ijms241511902_

Round 1

Reviewer 1 Report

The topic of the present paper „Effect of systemic inflammation in the CNS: A silent history of neuronal damage” is very interesting for readers. The present review aims to explain these processes and the pathophysiological interactions involved in encephalon damage without inflammatory cells or pathogens, the central nervous system being considered an immune-privileged tissue and is believed to be unable to initiate an inflammatory immune response in the presence of antigens.

The subsections of the manuscript are well described:

- Local infections generate systemic inflammatory response syndrome

- Endothelial damage   

- The central nervous system and systemic inflammation

- The circumventricular organs and systemic inflammation

- The central nervous system endothelium and systemic inflammation

- Glial cells and systemic inflammation

- Neurons and systemic inflammation

Based on the literature reviewed, the authors concluded that any systemic inflammatory event triggers neuroinflammation and leads to neuronal damage.

So, finally I conclude that:

-    the topic of the review is relevant and very interesting;

- the introduction provides sufficient background and includes relevant references;

-     the conclusions are consistent because evidence the presented arguments;

-     the reference list is variously relatively recently;

-      the manuscript is well written, and the text is easy to read.

Author Response

We sincerely appreciate your critical review of the manuscript and value the comments you have provided.

Reviewer 2 Report

The manuscript entitled “Effect of systemic inflammation in the CNS: A silent history of neuronal damage ” reviews the results of the studies performed in the field of neuroinflammation, in particular that caused indirectly by peripheral infections. Authors describe the physiopathogenic processes of neuronal damage induced by systemic inflammation and the neuroimmunological pathways involved.

The review is significant in the field of neuroinflammation and neurodegenerative diseases.

The manuscript  is well organized however there are some items that should be improved/corrected. They are listed below:

-          Abbreviations should be specified the first  time they appear in the text, including the abstract (BBB, BCSFB…). Moreover, there are abbreviations that have not been described through all the manuscript (i.e. CLP, BHLCR, UNV, …);

-          Abstract lines 36-40, the meaning of the sentence is not clear. I suggest changing with this: “Central nervous system (CNS) infections including meningitis and encephalitis, resulting from blood-borne spread of specific microorganisms, provoke nervous tissue damage due to the inflammatory process. Moreover, different pathologies such as sepsis can generate systemic inflammation. Bacterial LPS induces the release of inflammatory mediators and damage molecules, which are then released into the bloodstream and can interact with structures such as the CNS, thus modifying the BBB’s and BCSFB’s function and inducing aseptic neuroinflammation. During…”;

-          Abstract, line 46, since from the title through all the text authors talk about  central nervous system (CNS), the term “encephalon” (the area of central nervous system that includes all higher nervous centers, enclosed within the skull and continuous with the spinal cord, that is the brain) is confusing. I suggest to change “encephalon”  with “CNS”;

-          Page 2, line 84, change “The objective is…” with “Their function is…”;

-          Page 4, line 141, what “acute phase proteins ” refers to?, it should be specified;

Line 153, idem for “damaged molecules”;

Line 172, “…(34,000 +/- 76,000 172 pg/mL)…”, what does it mean?

-          Page 5, line 195-196, what “…The endothelium of blood vessels is composed of a monolayer of endothelial cells 195 resting in a basement membrane (appointments)…”, what appointments

-          Page 6, line 271, “occluding” instead of “occluding”;

-          Page 7, line 300, “HMGB1 protein” instead of “HMGB1 proteins“;

Line 321, “Its endothelium has specific characteristics to the rest of the endothelium” change with “Its endothelium has specific characteristics different to the rest of the endothelium”;

-          Page 10, lines 424-452, change “Systemic infectious interact with CNS, causing encephalitis and/or meningitis: however, it is sufficient that antigen such as LPS alters its vasculature.” With “Systemic infectious interact with CNS, causing encephalitis and/or meningitis. However, antigens such as LPS is sufficient to alter CNS’s vasculature.”;

Lines 426-427, change “BBB suffers intracellular molecular changes or alterations, but also structural and intercellular changes of its components.” with “BBB suffers not only intracellular molecular changes or alterations, but also structural and  intercellular changes of its components.”;

-          Page 11, line 443, change   “Active A1 from astrocytes was demonstrated…” with  “The induction of A1 reactive astrocytes was demonstrated…”;

Author Response

Thank you for your comments and insightful recommendations. All points have been addressed, and in the following list, you will find the page and line where the corrections were made. Regarding the specifications, the accompanying text provides an explanation, along with the page and line where the information was included:

- Abbreviations should be specified the first time they appear in the text, including the abstract (BBB, BCSFB…). Moreover, there are abbreviations that have not been described through all the manuscript (i.e. CLP, BHLCR, UNV, …);

The abbreviation terms were reviewed, and they are now indicated immediately after the first mention.

- Abstract lines 36-40, the meaning of the sentence is not clear. I suggest changing with this: “Central nervous system (CNS) infections including meningitis and encephalitis, resulting from blood-borne spread of specific microorganisms, provoke nervous tissue damage due to the inflammatory process. Moreover, different pathologies such as sepsis can generate systemic inflammation. Bacterial LPS induces the release of inflammatory mediators and damage molecules, which are then released into the bloodstream and can interact with structures such as the CNS, thus modifying the BBB’s and BCSFB’s function and inducing aseptic neuroinflammation. During…”;

The change has been made, and it is observed on page 1, line 36-42.

- Abstract, line 46, since from the title through all the text authors talk about central nervous system (CNS), the term “encephalon” (the area of central nervous system that includes all higher nervous centers, enclosed within the skull and continuous with the spinal cord, that is the brain) is confusing. I suggest to change “encephalon”  with “CNS”;

The change has been made, and it is observed on page 1, line 46.

-Page 2, line 84, change “The objective is…” with “Their function is…”;

The change has been made, and it is observed on page 2, line 87.

-Page 4, line 141, what “acute phase proteins ” refers to?, it should be specified;

The specified information has been provided and can be found on page 4, line 142.

Line 153, idem for “damaged molecules”;

The change has been made, and it is observed on page 4, line 155.

Line 172, “…(34,000 +/- 76,000 172 pg/mL)…”, what does it mean?

This means  “This molecule was elevated in 34 patients with severe sepsis and septic shock, and the serum concentration at 144 hrs, the last sampling point, was 300 times higher, than any of the other cytokines evaluated during the study” and it is observed on page 4, line 174.

-Page 5, line 195-196, what “…The endothelium of blood vessels is composed of a monolayer of endothelial cells 195 resting in a basement membrane (appointments)…”, what appointments

The change has been made, and it is observed on page 5, line 199.

- Page 6, line 271, “occluding” instead of “occluding”;

The change has been made, and it is observed on page 6, line 273.

- Page 7, line 300, “HMGB1 protein” instead of “HMGB1 proteins“;

The change has been made, and it is observed on page 7, line 302.

-Line 321, “Its endothelium has specific characteristics to the rest of the endothelium” change with “Its endothelium has specific characteristics different to the rest of the endothelium”;

The change has been made, and it is observed on page 7 , line 322 .

-Page 10, lines 424-452, change “Systemic infectious interact with CNS, causing encephalitis and/or meningitis: however, it is sufficient that antigen such as LPS alters its vasculature.” With “Systemic infectious interact with CNS, causing encephalitis and/or meningitis. However, antigens such as LPS is sufficient to alter CNS’s vasculature.”;

The change has been made, and it is observed on page 10, line 426-427.

-Lines 426-427, change “BBB suffers intracellular molecular changes or alterations, but also structural and intercellular changes of its components.” with “BBB suffers not only intracellular molecular changes or alterations, but also structural and  intercellular changes of its components.”;

The change has been made, and it is observed on page 10, line 428-429.

- Page 11, line 443, change   “Active A1 from astrocytes was demonstrated…” with  “The induction of A1 reactive astrocytes was demonstrated…”;

The change has been made, and it is observed on page 11, line 445.

Reviewer 3 Report

I have received the review article entitle “Effect of systemic inflammation in the CNS: A silent history of neuronal damage” by Millán-Solano et al., for evaluation. 

In this review, authors have reviewed the available literature describing the systemic inflammation in the CNS and its further consequences in brain. The information compiled in this review is timely and important. The review is timely, intuitive, and well written therefore, I have very few critics for this manuscript. 

My specific comments are:

1-    The review article is lacking the graphical summary abstract. I would suggest to authors to add a graphical abstract in the review.  

2-    The manuscript is missing few sections those are very important. I would suggest authors to add human clinical data on systemic inflammation and its impact on CNS under separate heading. 

3-    The review also missing the section describing the neurodegenerative disease caused systemic inflammation. I would suggest authors to incorporate this section under separate heading.

4-    In the last, I would suggest authors to add future prospective and current diagnostic methods that being used to diagnose the effect of systemic inflammation on brain.

 Minor English corrections.

Author Response

Thank you for your comments. The sections you suggested including have been worked on and incorporated into the text. Below, you will find the title of each section and its position in the document. Additionally, the graphical abstract has been included.

1-    The review article is lacking the graphical summary abstract. I would suggest to authors to add a graphical abstract in the review.  

The graphical abstract has been included, providing a concise summary of the events described in the manuscript and following the editorial guidelines.

2-    The manuscript is missing few sections those are very important. I would suggest authors to add human clinical data on systemic inflammation and its impact on CNS under separate heading. 

The requested section titled: “Clinical data of sepsis and neurological damage” has been included and can be found on page 15, line 650-675.

3-    The review also missing the section describing the neurodegenerative disease caused systemic inflammation. I would suggest authors to incorporate this section under separate heading.

The requested section titled:_ 10. “Systemic inflammation and neurodegeneration “ has been included and can be found on page 16, line 702-759.

4-    In the last, I would suggest authors to add future prospective and current diagnostic methods that being used to diagnose the effect of systemic inflammation on brain.

The requested section titled: “Clinical data of sepsis and neurological damage”_has been included and can be found on page 15, line 676-699.

Round 2

Reviewer 3 Report

Congratulation.

Minor issues